# Possible Beneficial Actions of Caffeine in SARS-CoV-2

**DOI:** 10.3390/ijms22115460

**Published:** 2021-05-22

**Authors:** Bianca S. Romero-Martínez, Luis M. Montaño, Héctor Solís-Chagoyán, Bettina Sommer, Gemma Lizbeth Ramírez-Salinas, Gloria E. Pérez-Figueroa, Edgar Flores-Soto

**Affiliations:** 1Departamento de Farmacología, Facultad de Medicina, Universidad Nacional Autónoma de México, CDMX CP 04510, Mexico; biancasromero_@hotmail.com (B.S.R.-M.); lmmr@unam.mx (L.M.M.); 2Laboratorio de Neurofarmacología, Instituto Nacional de Psiquiatría Ramón de la Fuente Muñiz, CDMX CP 14370, Mexico; hecsolch@imp.edu.mx; 3Laboratorio de Hiperreactividad Bronquial, Instituto Nacional de Enfermedades Respiratorias “Ismael Cosío Villegas”, CDMX CP 14080, Mexico; bsommerc@hotmail.com; 4Laboratorio de Diseño y Desarrollo de Nuevos Fármacos e Innovación Biotécnológica (Laboratory for the Design and Development of New Drugs and Biotechnological Innovation), Escuela Superior de Medicina, Instituto Politécnico Nacional, CDMX CP 11340, Mexico; gemali86@hotmail.com; 5Laboratorio de Investigación en Inmunología y Proteómica, Hospital Infantil de México Federico Gómez, CDMX CP 06720, Mexico; gera.pfi3@gmail.com

**Keywords:** caffeine, COVID-19, SARS-CoV-2, airway smooth muscle, immunomodulatory effects, antiviral activity

## Abstract

The COVID-19 pandemic has established an unparalleled necessity to rapidly find effective treatments for the illness; unfortunately, no specific treatment has been found yet. As this is a new emerging chaotic situation, already existing drugs have been suggested to ameliorate the infection of SARS-CoV-2. The consumption of caffeine has been suggested primarily because it improves exercise performance, reduces fatigue, and increases wakefulness and awareness. Caffeine has been proven to be an effective anti-inflammatory and immunomodulator. In airway smooth muscle, it has bronchodilator effects mainly due to its activity as a phosphodiesterase inhibitor and adenosine receptor antagonist. In addition, a recent published document has suggested the potential antiviral activity of this drug using in silico molecular dynamics and molecular docking; in this regard, caffeine might block the viral entrance into host cells by inhibiting the formation of a receptor-binding domain and the angiotensin-converting enzyme complex and, additionally, might reduce viral replication by the inhibition of the activity of 3-chymotrypsin-like proteases. Here, we discuss how caffeine through certain mechanisms of action could be beneficial in SARS-CoV-2. Nevertheless, further studies are required for validation through in vitro and in vivo models.

## 1. Introduction

In December 2019, a series of unexplained cases of atypical pneumonia were reported in Wuhan, China, with high transmission. This disease, which was subsequently named coronavirus disease 2019 (COVID-19), has spread rapidly worldwide, affecting a large part of the human population [1,2]. The World Health Organization (WHO) officially named the virus severe acute respiratory syndrome coronavirus 2 (SARS-CoV-2) [3]. SARS-CoV-2 belongs to the coronavirus family, a group of enveloped, single-stranded, positive-sense, RNA genome viruses. The virion contains four main structural proteins: the nucleocapsid (N) located in the nucleocapsid and in the viral envelope we can find the spike (S), membrane (M), and envelope (E) proteins. The S protein has been determined to facilitate viral entry into the host cell; this occurs through the complex formed by a receptor-binding domain (RBD; a subunit of the S protein) and the angiotensin-converting enzyme 2 (ACE2) found in the membrane of the host cell, mainly pneumocyte type II cells [1,4,5,6]. The number of patients is increasing day by day around the world, but some infected patients are asymptomatic or experience a mild disease course (fever, cough, chest tightness, dyspnea, etc.). However, patients with severe symptoms may present severe respiratory tract infections, severe pneumonia, acute respiratory distress syndrome (ARDS), multiple organ failure, and death [7,8,9,10,11]. This is because the virus induces aberrant host immune responses, and modulation of the host immune response is the key to fight SARS-CoV-2 [12]. Among the theories, it is hypothesized that SARS-CoV-2 damages tissues due to the deterioration of inflammation mechanisms and cytokine storms given the pathophysiology of SARS-CoV-2 [13,14]. The WHO reported that as of 28 February 2021 there had been approximately 113,864,015 cases of COVID-19 and that the number of deaths had been approximately 2,526,793 worldwide. In Mexico, to this date there had been a total of 2,084,128 positive cases with a total of 185,257 deaths, and, unfortunately at the time of writing, the number of COVID-19 positive cases is increasing daily [15].

Many researchers, through their respective research areas, are facing and directing contributions to overcome the COVID-19 pandemic. Although the development and use of vaccines (with high efficiency) has been the first option, at the time of writing, only 0.67% of the world’s population, according to the WHO, has been vaccinated. Therefore, current treatments are largely directed toward symptom management and vital support in severe cases [16]. Given the necessity for efficient therapeutic options in the intervention of SARS-CoV-2, two main routes in the drug discovery process for a viral infection are being undertaken: the discovery or synthesis of a new effective drug and the repurposing of an already existing drug. Drug discovery is a laborious process, usually taking several years and at large expense; therefore, taking into consideration the urgency arising from the pandemic currently plaguing our world, the option to repurpose already existing drugs or pharmacological compounds seems the more feasible option [17].

Until now, the therapeutic options to contain the COVID-19 pandemic have been based on prevention of transmission, detection of travelers, and public healthcare measures [17]. There is no effective treatment for SARS-CoV-2 infection and the drugs mainly used include antiviral protease inhibitors that impede the viral replication of SARS-CoV-2 by inactivating proteases essential for the replication [18]. The identification of specific drug targets that inhibit the life cycle of SARS-CoV-2 still require further investigation.

Another strategy is to use inflammation inhibitors, because experimental and clinical tests have shown that the damage caused by the virus is related to an altered inflammatory response and, in some patients, to an abnormal release of pro-inflammatory cytokines [18]. Low molecular heparins, plasma, and hyperimmune immunoglobulins are also utilized on a case-by-case basis, generally in severe COVID-19 patients, and are used to mitigate the complications and sequelae that can arise from the infection [18,19,20]. Although a variety of therapies may be a short-term strategy to deal with COVID-19, there is still an obvious lack of specific treatment for the disease [7]. Even if these anti-inflammatory agents are a promising line of treatment, they still require further study through randomized clinical trials.

Caffeine (1,3,7-trimethylxanthine) is a methylxanthine alkaloid found in the seed, fruit, and leaves of a variety of plants native to Africa, Southeast Asia, and South America. In addition to coffee and tea, it can also be found in cocoa beans, yerba mate leaves (used to make herbal teas), and guarana berries (used in various beverages and supplements). It is a common stimulant that is consumed daily around the world [21]; it can be synthesized and added to beverages and foods, including soft drinks, beverages, and tablets in a wide variety of over-the-counter formulations, such as combined diet aids and pain relievers. If consumed within the recommended dose (400 mg/day for adults), its most widely sought-after effect is as a mild stimulant of the central nervous system (CNS), due to its capacity to cross with ease the blood–brain barrier, which can cause a reduction in fatigue and increase wakefulness and awareness [22,23]. Caffeine is used successfully in the treatment of apnea of prematurity [24], exerting its antagonism on the adenosine receptors in the respiratory centers of the brainstem [25]. In chronic lung disease of prematurity, it inhibits non-selective phosphodiesterase and increases cyclic adenosine monophosphate levels (cAMP), directly relaxing the pulmonary vascular muscle of the baby and improving its oxygenation [26]. In asthma, caffeine has been utilized in adults with mild asthma, showing a moderate improvement in lung function when low doses of 5 mg/kg of body weight are administered [27]. There are reports where people with exercise-induced bronchoconstriction (EIB) could alleviate it by ingesting caffeine before exercise (7 mg/kg) [28]. The mechanisms proposed for the bronchodilator effect are mostly through its activity as an inhibitor of phosphodiesterase and antagonism of the adenosine receptor [29]. It should be mentioned that consumed caffeine is easily distributed throughout the body and permeates through the cellular membrane due to its amphiphilic properties [30]. In the liver, it is metabolized by cytochrome P-450 (CYP) enzymes and biotransformed by CYP1A2 in three active metabolites, mainly paraxanthine (81.5%), theobromine (10.8%), and theophylline (5.4%), which are excreted in the urine [31]. A moderate consumption of caffeine is usually considered safe and acute toxicity is rare, but some of the symptomology that can be present are nausea, headaches, insomnia, nervousness, tachycardia, arrhythmias, gastrointestinal disturbances, and seizures. The consumption of caffeine in combination with alcohol could even result in death [23,32]. These could be the possible side effects if caffeine were to be used for treating COVID-19 patients.

We and other research groups have studied the pharmacological activities of caffeine in the smooth muscle of the respiratory tract, including mainly adenosine receptor antagonism, phosphodiesterase inhibition, and intracellular calcium release from the sarcoplasmic reticulum (SR), and as a taste receptor type 2 ligand (TAS2R) [33,34,35,36]. It is important to emphasize that these mechanisms are found in almost all tissues and cells of the body. Recently, caffeine has taken on an essential role in the fight against SARS-CoV-2 since it can play a role in the defense against this virus [6,7,36]. In this review, we analyze our experience in the handling of caffeine and the analysis of new experimental data that may have health benefits against SARS-CoV-2 in multiple organ systems, via inactivation of the virus, blocking the viral binding with ACE2, along with its immunomodulatory and anti-inflammatory roles, as well as benefits in patients with COVID-19, especially as caffeine is consumed daily.

## 2. Pharmacological Mechanisms of Caffeine in Airway Smooth Muscle

Recently, Kalidhindi et al., using confocal imaging, found that ACE2 is expressed in human airway smooth muscle (ASM), although an entry of SARS-CoV-2 has not been demonstrated [37]. The importance of these cells lies in the possibility that SARS-CoV-2 targets and damages the epithelial layers with subsequent influences on the underlying mesenchymal cells (ASM, fibroblasts), leading to an alteration in the reactivity of the airways, inflammation, and long-term fibrosis. In this context, SARS-CoV-2 can directly influence the functionality of multiple signaling pathways in lung cells, including ASM [37].

### 2.1. Adenosine Receptors

In the 1980s, it was found that adenosine administered by inhalation induces bronchoconstriction in subjects with asthma [38] and increases the concentrations of pro-inflammatory mediators such as histamine, tryptase, leukotrienes, and prostaglandins discharged from mast cells [39,40]. These findings pointed out that adenosine may result in bronchoconstriction through mast cell activation [41,42]. In asthmatics, adenosine concentrations are elevated in bronchoalveolar lavage fluid (BAL) [43] and in exhaled air condensate [44,45]. In conclusion, adenosine produces bronchoconstriction, inflammation, and mucus, which lead to airway obstruction. In addition, it may function as a paracrine mediator of the inflammatory response of the lung, especially in the allergic-asthma phenotype [38,46,47]. The effects of adenosine are exerted through G-protein-coupled receptors (GPCRs) and they have been classified into four subtypes, namely, adenosine receptors A_1_, A_2A_, A_2B_, and A_3_ [48]. When activated, these receptors increase or decrease the concentration of cAMP; subtypes A_1_ and A_3_ are coupled to Gi proteins that decrease the intracellular concentration of cAMP, while subtypes A_2A_ and A_2B_ are coupled to Gs proteins and can increase the intracellular concentration of cAMP [49] (Figure 1).

The IC_50_ of caffeine has been reported in neuronal tissues to be 160–210 μM for A_2B_ receptors and 17 nM for A_2A_ receptors [50]. The adenosine receptor pathways are useful targets in asthma treatment through the effects observed in airway smooth muscle (ASM) or on various types of immune and non-immune cells [51]. Adenosine receptors have been reported to be expressed in most immune cells and participate in the regulation of various putative functions. A_2A_ is the most widely expressed, particularly in asthmatic patients, with the A_1_, A_2B_, and A_3_ subtypes being expressed only in certain cell types [52]. Inhibition could lead to more refined asthma management. The selectivity of caffeine analogs for A_2A_ receptors has been found to be somewhat increased by the replacement of one or two methyl groups of caffeine with a propyl or propargyl substituent [53]. Indeed, 3,7-dimethyl-1-propargylxanthine (DMPX) has been successfully used as a selective A_2_ antagonist [54]. On the other hand, caffeine-inspired non-xanthine heterocyclic antagonists for A_2B_ receptors have also been developed, including a variety of 9-deaza-xanthines and 9-deaza-adenines [55,56].

### 2.2. Cyclic Nucleotide Phosphodiesterases

There are 11 families of phosphodiesterase (PDE) genes (PDE1–PDE11), which encode the PDE proteins that hydrolyze cyclic adenosine monophosphate (cAMP) and cyclic guanosine monophosphate (cGMP) to inactive 5′ AMP and 5′ GMP, respectively, regulating their intracellular levels [57]. These cyclic nucleotides activate a variety of cellular processes including the relaxation of airway smooth muscle and the release of inflammatory mediators [58] (Figure 1). PDE inhibition represents a potential mechanism to modulate airway smooth muscle contraction and the release of inflammatory mediators [57]. PDE3 and PDE4 are located in the tracheal and vascular smooth muscle, and PDE4 has a wide distribution in tissues including brain, gastrointestinal (GI) tract, spleen, lung, heart, testis, and kidney [58]. Furthermore, PDE4 is expressed in almost all inflammatory cell types except platelets that contain PDE3 and PDE5 [57]. The IC_50_ of caffeine to inhibit phosphodiesterase ranges from 500 μM to 1 mM [59], while theophylline is considered a more potent PDE4 inhibitor due to a lower IC_50_ of 70 μM. Theophylline is considered a more potent PDE4 inhibitor because cAMP values of 40% equivalent to 18 μg/mL have been found in plasma levels; at this concentration, relaxation of the bronchial tissue is induced [60]. Recently, Sildenafil, an approved PDE5 inhibitor used in the treatment of erectile dysfunction and pulmonary hypertension [61] that improves pulmonary hemodynamics by reducing vascular resistance in idiopathic pulmonary fibrosis [62,63], has proven to be effective in the treatment of COVID-19 [64]. We believe that caffeine, through its PDE inhibitory capacity, could potentially have similar effects, although more research is needed in this regard.

### 2.3. Agonist of the Type 2 Taste Receptor

In ASM, the canonical signaling pathway for the type 2 taste receptor (TAS2R) is through the coupling of a bitter compound to a GPCR receptor, which activates the Gqα subunit gustducin. While Gβ3 and Gγ13 form a complex, the subunit Gα gustducin dissociates from the heterodimer Gβ3/Gγ13, which activates the phospholipase C-β_2_ (PLC-β_2_) responsible for hydrolyzing phosphatidylinositol-4,5-bisphosphate in inositol-1,4,5-triphosphate (IP_3_) and diacylglycerol. The IP_3_ will bind to the IP_3_ receptor (IP3R) on the SR and allow the Ca^2+^ to be released (Figure 1). It has been proposed that this released Ca^2+^ induces the opening of high-conductance calcium-activated potassium channels (BK_Ca_), which are responsible for the hyperpolarization of the membrane and the relaxation of ASM [61,62,63,64,65]. In ASM, the TAS2R subtypes that are expressed are 7, 10, 14, 43, and 46, and caffeine is a known agonist of the TAS2R receptors [62]. It has been seen that in ASM, caffeine reverses the agonist-induced bronchoconstriction by blocking Ca^2+^ oscillations, decreasing the sensitivity to Ca^2+^ via the inhibition of released Ca^2+^ by the IP3R [66]. In this context, caffeine may facilitate breathing by contributing to widening the airway caliber, although much more studies are needed.

## 3. Immunomodulatory Effects of Caffeine

Among the many possible benefits caffeine might indirectly have in the infection caused by SARS-CoV-2, a strong emphasis should be made in the immunomodulatory effects it possesses [34,67]. The application of a bolus of caffeine at a dose of 6 mg/kg produces an increase in the total lymphocyte and CD8+ lymphocyte count [68]. Exposure of human cells to high and prolonged concentrations of caffeine has been shown to promote NK cell activity [69,70]. NK cells and lymphocytes are among the first cells activated against a viral threat and play a crucial role in the pathophysiology of COVID-19 infection (Figure 2).

### 3.1. Adenosine Receptors

Some caffeine immunomodulatory effects can be attributed to its direct activity on immune cells, many of which express adenosine receptors, and caffeine is a well-known antagonist of these receptors. The stimulation of A_1_ has been shown to promote the pro-inflammatory functions of neutrophils and eosinophils and promote monocyte phagocytosis, dendritic cell chemotaxis, and mucus production [52]. On the other hand, A_2A_ stimulation has been shown to inhibit degranulation from mast cells and neutrophils and neutrophil adherence to the endothelium; its activation has also been shown to inhibit IL-12 and TNF-α secretion from monocytes and macrophages and inhibits IL-6 and IL-8 secretion from endothelial cells [52].

On the other hand, A_2B_ stimulation promotes a pro-inflammatory state: in mast cells, it induces degranulation, cytokine secretion, and IgE synthesis, whereas in epithelial cells, it promotes IL-19 secretion. Moreover, VEGF and IL-8 secretion is promoted in endothelial cells, and IL-16 is secreted from bronchial smooth muscle and fibroblasts [52]. Little is known about the A_3_ receptor functions, but its activity has been shown to be mainly inhibiting degranulation in neutrophils and inhibiting degranulation and chemotaxis in eosinophils.

It has been reported that adenosine plays a key role in regulating pulmonary inflammation and repair [71,72]. Remarkably, in acute inflammation, it significantly lowers cytokine production, immune cell migration, and vascular permeability, while during chronic inflammation, continuous, sustained adenosine stimulation increases cytokine secretion and immune cell infiltration into the lungs [73]. Interestingly, caffeine administration has been shown to partially revert the consequences of persistent adenosine stimulation [74], adding another interesting feature to the caffeine anti-inflammatory spectrum [75].

Thus, the participation of adenosine receptor pathways in the development of the cytokine storm in COVID-19 patients might be diminished through the protective effects of caffeine, limiting lung inflammation [52] (Figure 2).

### 3.2. Agonist of the Type 2 Taste Receptor

Anosmia and ageusia are characteristic symptoms of viral respiratory infections, particularly in that caused by SARS-CoV-2. TAS2R receptors are found to be expressed in many cell types, not only chemosensory epithelial cells but also throughout the airway and other systems, including leukocytes, mast cells, neutrophils, monocytes, eosinophils, and macrophages [64,76]. TAS2R agonists have demonstrated the inhibition of allergen-induced airway inflammation, inhibition of LPS-induced cytokine release, remodeling, and hyper-responsiveness [64,76]. Caffeine, being a known agonist of these receptors, could participate in accelerating the recovery of the sense of smell and could potentially contribute to mitigating the severity of the infection caused by SARS-CoV-2 due to the immunomodulatory effects attributed to these receptors [12,61]. In the first line of defense in the upper respiratory system against viral pathogens, the administration of caffeine has been demonstrated to modify certain immunological elements found in saliva, increasing the secretion of immunoglobulins and serum albumin, as well as lowering the levels of cystatin SN, a known inhibitor of protease [12,77]. In addition, the activation of TAS2Rs in epithelial cells of the upper respiratory tract has been shown to promote the secretion of antimicrobial peptides (AMPs), NO, and H_2_O_2_, which directly target viruses [78,79].

In macrophages, the activation of TAS2R has been shown to promote phagocytosis through NO and cAMP pathways, as well as the inhibition of TNF-α, CCL3, and CXCL8 production induced by LPS [61,64,80]. TAS2R agonists also inhibit pro-inflammatory cytokine production from leukocytes and IgE-mediated histamine and prostaglandin D_2_ released by mast cells [61,81,82]. In a murine model, chronic or high acute doses of caffeine treatment protected against lung injury by lowering neutrophil recruitment and inhibiting secretion of TNF-α and IL-1 in an A_2A_-independent manner but mediated through a cAMP pathway, possibly through the activation of TAS2R. However, a deleterious effect was observed when low acute doses of caffeine were administered, and this effect might be explained by the inhibition of the anti-inflammatory pathways mediated by the A_2A_ [83].

Similar results were observed in a previous study, where caffeine suppressed TNF-α plasma levels through the cAMP/PKA pathways, possibly through the activation of a TAS2R receptor [84] (Figure 2). Seemingly, caffeine can act as an immunosuppressor by reversing the overexpression of cytokines via TAS2R activation.

## 4. Probable Antiviral Activity of Caffeine in SARS-CoV-2 Infection

One of the first mechanisms that initiates the innate immune response is inflammasome NOD-like receptor 3 (NLRP3), an intracellular pattern recognition receptor (PRR) that recognizes pathogen-associated molecular patterns (PAMPs) and danger-associated molecular patterns (DAMPs) [85]. Caffeine has been shown to reduce its expression as well as suppress its activation through the mitogen-activated protein kinase (MAPK)/NF-κB signaling pathway. Consequently, the production of IL-1β and IL-18 is inhibited [67,86,87]. Notably, caffeine has also been observed to inhibit viral RNA synthesis and viral protein synthesis by some virus species such as the Newcastle disease virus, influenza virus, poliovirus, herpes simplex virus type 1 (HSV-1), human immunodeficiency virus (HIV), vaccinia virus, and polyomavirus [67,88,89,90,91,92]. In particular, an in vitro study showed that caffeine was able to inhibit the replication of the hepatitis C virus [67,93] (Figure 2).

### 4.1. Caffeine Might Be an Inhibitor of the RBD–ACE2 Complex

A key strategy in drug discovery against COVID-19 has been targeting the viral entrance into the host cell. Recently, Mohammadi et al. characterized the viral entrance through the formation of a complex between the RBD found in the S protein of the virus and the ACE2, the functional receptor for SARS-CoV-2 in the host cell’s membrane. In a recent in silico study, caffeine was proposed as an effective inhibitor of the complex, both alone and in combination with antiviral agents [94].

The interactions between caffeine and caffeine in combination with antiviral agents with the RBD/ACE2 complex were evaluated through molecular dynamic (MD) simulation and molecular docking. The two principal epitopes selected from the S protein crystal structure were recognized in the Protein Data Bank (PDB ID: 6VW1 and 6LZG). In addition, the interaction energies (IEs) were calculated between the RBD–ACE2 complex and the drug. From the data generated in MDs, the IE between caffeine and RBD–ACE2 complexes was calculated, showing a strong interaction with 6LZG. The results show that caffeine is especially efficient at interacting with 6LZG, thus blocking the formation of the RBD–ACE2 complex [94].

The possibility of using a combination of caffeine and antiviral agents was also explored through MD simulations and molecular docking, and it was shown that caffeine in combination with ribavirin has a synergic interaction in blocking the 6VW1 site, with a binding free energy of −6.76 (kcal/mol) and IE of −2000 (kcal/mol). Additionally, in the case of the 6VW1 complex, caffeine with favipiravir and ribavirin forms a more efficient structure against SARS-CoV-2 in terms of non-binding interaction energy, demonstrating a stable and promising binding tendency of caffeine with ACE2 and consequently possible inhibition of ACE2 against SARS-CoV-2 [94].

The most suitable interaction combination with ACE2 in absence of the S protein was also addressed. Among the various antiviral agents plus caffeine combinations explored, the strongest interaction was with ACE2 + remdesivir + caffeine with an IE value of −396.68 (kcal/mol). This negative value indicates that the complex is stable, preventing the formation of the S protein–ACE2 complex, and thus preventing the SARS-CoV-2 virus from infecting cells and continuing its viral cycle. Considering these results, the potential for the application of caffeine, both alone and in combination with antiviral agents, can be noticed, even more so since Tong et al. reported that ribavirin did not provide a survival benefit in comparison with the control treatment (involving only supportive therapy). As such, its in vivo activity against SARS-CoV-2 requires further investigation [95] (Figure 2).

### 4.2. Inhibition of 3-Chymotrypsin-Like Protease

One of the mechanisms necessary for the viral transcription and replication of SARS-CoV-2 depends on two polyproteins involved in the release of the functional polypeptide. Releasing these polypeptides requires the participation of 3-chymotrypsin-like protease (3CL_pro_). Considering the importance of this protease, it has been investigated as a potential pharmacological target, searching for both existing drugs and natural compounds that could inhibit its activity [96,97]. Caffeine, other methylxanthines, and structurally similar compounds that possess a 4-pyridone ring in their structure have demonstrated an inhibitory effect on 3CL_pro_ through simulations of molecular dynamics and energy calculations [95]. The molecular docking between the 3CL_pro_ protein and caffeine shows an affinity of −5.6 kcal/mol and also forms hydrogen bonds with amino acids Cys145 and Glu166. These residues are key in the inhibition of the protease activity. This would prevent the generation of non-structural proteins that are essential for viral replication and thus the formation of new virions.

Finally, MD simulations were performed for the complexes formed in the dockings of the protein 3CL_pro_–caffeine and other similar compounds; these simulations were 200 ns [95]. The root-mean-square fluctuation (RMSF) values of the molecular dynamics of the 3CL_pro_–caffeine complexes were obtained; they demonstrated that the amino acids of the active site remained in stable conformation during simulation. Therefore, caffeine is recommended as a possible inhibitor of 3CL_pro_, further pointing out that this alkaloid has the potential to become a new therapeutic agent; however, further in vitro and in vivo studies are recommended to corroborate these findings [96] (Figure 2).

## 5. Conclusions

Despite the shared interests worldwide to provide effective therapeutic options against SARS-CoV-2, no specific antiviral treatment exists, and the primary options are based on symptomology treatment and vital support. In this review, we ascertained the possible health benefits that caffeine might provide, both directly and indirectly in terms of SARS-CoV-2 infection, by favoring bronchodilatation and immunomodulation and probably by hindering viral intracellular transcription.

## Figures and Tables

**Figure 1 ijms-22-05460-f001:**
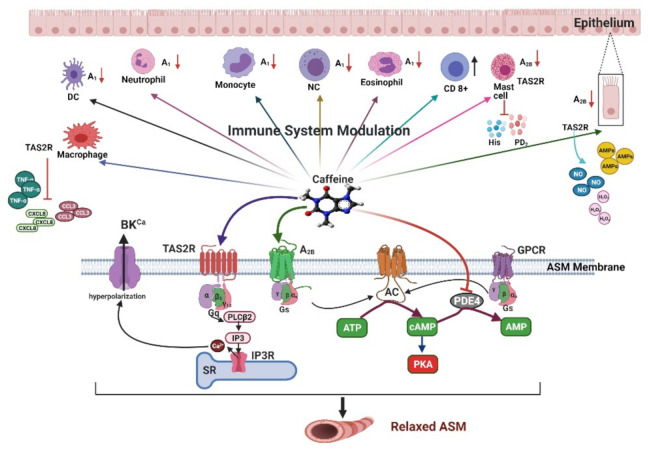
Schematic diagram of the immunomodulatory and bronchodilatory effects of caffeine. Various cells of the immune system express adenosine receptors (ARs) and TAS2R receptors that can modulate their activity; caffeine is a known antagonist of the AR receptor and an agonist of TAS2R receptors. Antagonism of A_1_ and A_2B_ on immune cells will decrease pro-inflammatory activity. Caffeine activation of TAS2R in epithelial cells will increase the secretion of antimicrobial peptides (AMPs), NO, and H_2_O_2_, all of which have direct effects on the virus. In macrophages it inhibits pro-inflammatory cytokine production, and in mast cells it inhibits histamine and prostaglandin D_2_ release. In ASM, caffeine can activate various pathways that promote bronchodilation: it activates TAS2R and A_2_ receptors and inhibits PDE4 to induce relaxation. At high concentrations, it can also increase the open probability of RyR; this last mechanism has no therapeutic use. ASM, airway smooth muscle; AR, adenosine receptor; DC, dendritic cell; GPCRs, G-protein-coupled receptors; TAS2R, type 2 taste receptor; NC, natural killer cell; CD, lymphocyte CD 8+; His, histamine; PD2, prostaglandin D2; PDE4: phosphodiesterase 4; AMPs, antimicrobial peptides; IP_3_, inositol 1,4,5-triphosphate; PLC-β_2_, phospholipase C-β_2_; SR, sarcoplasmic reticulum; PKA, protein kinase A; cAMP, cyclic adenosine monophosphate; GPCR, G-protein-coupled receptor; IP3R, IP_3_ receptor.

**Figure 2 ijms-22-05460-f002:**
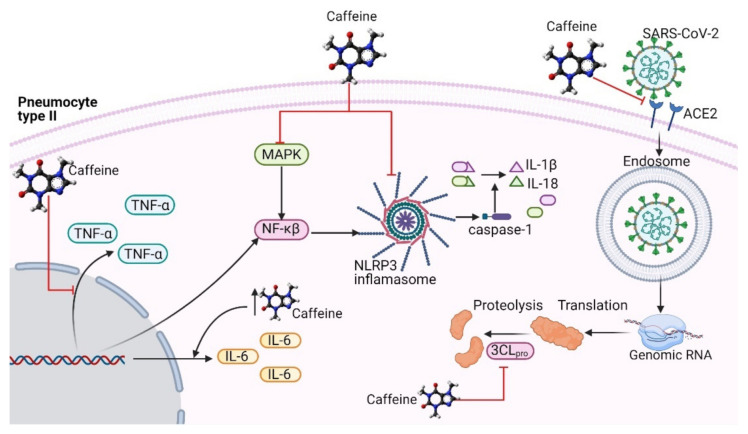
Schematic diagram of the proposed antiviral mechanisms of caffeine against SARS-CoV-2. Caffeine can inhibit the production of TNF-α and the expression of the inflammasome NLRP3 and its activity by the MAPK/NF-κβ pathway, decreasing the production of IL-1β and IL-18. Caffeine also inhibits viral entrance by blocking the RBD and ACE2 complex formation. Caffeine also inhibits 3CL_pro_, a protease required to release two polypeptides that are necessary for viral transcription and replication. TNF-α, tumor necrosis factor alpha; NF-κβ, nuclear factor kappa beta; ACE2, angiotensin-converting enzyme 2; NLRP3, inflammasome NOD-like receptor 3; IL-6, interleukin 6; IL-1β, interleukin 1 beta; IL-18, interleukin 18; MAPK, mitogen-activated protein kinase; 3CL_pro_, 3-chymotrypsin-like protease.

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
