# Peer review of "Possible Beneficial Actions of Caffeine in SARS-CoV-2"

_ijms, 2021, doi:10.3390/ijms22115460_

Round 1
Reviewer 1 Report
The review article is original, well written and appealing. Perhaps some part could be shortned (e.g. the introduction arrive to the topic too slowly). I'd add a paragraph about the pharmacological issues of a possible therapy with caffeine in patients with COVID-19.
Author Response
The review article is original, well written and appealing. Perhaps some part could be shortned (e.g. the introduction arrive to the topic too slowly). I'd add a paragraph about the pharmacological issues of a possible therapy with caffeine in patients with COVID-19.
Answer: Thank you for your observation!
In accordance with your observation and one from the second reviewer regarding the same issue, we modified our focus from a clinical approach to a basic science level, because we agree that caffeine research is not enough yet to propose this alkaloid as a COVID-19 treatment. The side effects of caffeine consumption have been added to the text. Hypothetically, the use of caffeine in COVID-19 patients could have similar side effects to those already described in the literature, especially in high acute doses that can produce toxicity.
The paragraph added to the amended text reads as follows: “The moderate consumption of caffeine is usually considered safe and acute toxicity is rare, but some of the symptomology that can be present are nausea, headaches, insomnia, nervousness, tachycardia, arrhythmias, gastrointestinal disturbances, and seizures. The consumption of caffeine in combination with alcohol could even result in death. [23, 32]. The aforementioned symptoms might develop as side effects if caffeine were used in COVID-19 patients”. Page 3 Line 116-120
Reviewer 2 Report
The presented review summarizes the mechanisms of caffeine action, that could be potentially effective against SARS-CoV-2, thus enabling the application of caffeine in the future anti-COVID-19 treatment.
The ability of caffeine to modulate the immune response via the decreasing the expression of NLRP3, as well as supressing its activation through MAPK/NF-kB signalling pathway, could result in the ameliorating the course of cytokine storm, which is the most life-threatening complication of COVID-19 disease. Moreover the potential of caffeine to inhibit the formation of complex between the RBD fragment of viral S protein and ACE2 receptor, could block SARS-CoV-2 entrance to host cells. The third important mechanism of the caffeine action, discussed by the authors, is the inhibition of 3-chymotrypsin-like protease, the enzyme that is crucial for viral replication within the infected host cell.
The experiments that enabled to demonstrate such caffeine properties have been based on molecular dynamic (MD) stimulation and molecular docking. The results of such experiments should not be, in my opinion, directly translated into clinical medicine.
The authors propose for future studies the combination of certain antiviral drugs and caffeine. Most antiviral compounds, listed by author, such as ribavirin, lopinavir-ritonavir have been already excluded from clinical trials as being ineffective against SARS-CoV-2.
As we lack the evidence of very early phase of SARS-CoV-2 infection, it would be difficult to interact with viral entrance to the host cells.
The inhibition of viral replication by caffeine could be theoretically used for enhancement of antiviral activity of other agents. The results of clinical trials documented that antiviral treatment is of benefit only in the first week of symptomatic SARS-CoV-2 disease. The only drug that showed the antiviral potential, in this indication, was remdesivir.
In developed cytokine storm, the anti-inflammatory treatment with IL-6 receptor antagonist (tocilizumab) and dexamethasone proved to decrease mortality of SARS-CoV-2. Any new compound tested in this phase of COVID-19 disease should have strong recommendation from phase I and phase II clinical studies.
Therefore, in my opinion, the authors should be aware of the fact that the demonstration of certain mechanisms of caffeine action, doesn’t directly translate into its possible use in clinical medicine. Likewise, despite many reports concerning antiviral properties of caffeine published in early twenties, no indications for therapeutic caffeine use in viral diseases, have been proposed, until now. As such, the conclusions concerning clinical use of caffeine in SARS-CoV-2 should not be presented until we have more clinical evidence of the effectiveness of this compound.
In summary, I would recommend to publish the article as more scientific point of view, than clinical review. The parts describing the current status of anti - SARS-CoV-2 therapy either have to be actualized according to current evidence of knowledge, or deleted.
Author Response
The presented review summarizes the mechanisms of caffeine action, that could be potentially effective against SARS-CoV-2, thus enabling the application of caffeine in the future anti-COVID-19 treatment.
The ability of caffeine to modulate the immune response via the decreasing the expression of NLRP3, as well as supressing its activation through MAPK/NF-kB signalling pathway, could result in the ameliorating the course of cytokine storm, which is the most life-threatening complication of COVID-19 disease. Moreover the potential of caffeine to inhibit the formation of complex between the RBD fragment of viral S protein and ACE2 receptor, could block SARS-CoV-2 entrance to host cells. The third important mechanism of the caffeine action, discussed by the authors, is the inhibition of 3-chymotrypsin-like protease, the enzyme that is crucial for viral replication within the infected host cell.
The experiments that enabled to demonstrate such caffeine properties have been based on molecular dynamic (MD) stimulation and molecular docking. The results of such experiments should not be, in my opinion, directly translated into clinical medicine.
The authors propose for future studies the combination of certain antiviral drugs and caffeine. Most antiviral compounds, listed by author, such as ribavirin, lopinavir-ritonavir have been already excluded from clinical trials as being ineffective against SARS-CoV-2.
As we lack the evidence of very early phase of SARS-CoV-2 infection, it would be difficult to interact with viral entrance to the host cells.
The inhibition of viral replication by caffeine could be theoretically used for enhancement of antiviral activity of other agents. The results of clinical trials documented that antiviral treatment is of benefit only in the first week of symptomatic SARS-CoV-2 disease. The only drug that showed the antiviral potential, in this indication, was remdesivir.
In developed cytokine storm, the anti-inflammatory treatment with IL-6 receptor antagonist (tocilizumab) and dexamethasone proved to decrease mortality of SARS-CoV-2. Any new compound tested in this phase of COVID-19 disease should have strong recommendation from phase I and phase II clinical studies.
Therefore, in my opinion, the authors should be aware of the fact that the demonstration of certain mechanisms of caffeine action, doesn’t directly translate into its possible use in clinical medicine. Likewise, despite many reports concerning antiviral properties of caffeine published in early twenties, no indications for therapeutic caffeine use in viral diseases, have been proposed, until now. As such, the conclusions concerning clinical use of caffeine in SARS-CoV-2 should not be presented until we have more clinical evidence of the effectiveness of this compound.
In summary, I would recommend to publish the article as more scientific point of view, than clinical review. The parts describing the current status of anti - SARS-CoV-2 therapy either have to be actualized according to current evidence of knowledge, or deleted.
Answer: We appreciate your insight!
The enrichment of our work through the various observations highlighted in your response lead us to change the focus of the work from a clinical proposal to a basic research perspective. We decided to modify the title to “Possible benefic actions of caffeine in SARS-CoV-2”. Considering your accurately mentioned statement, the inclusion of the drug therapies already studied in SARS-CoV-2 distracted from the original direction of the work and we decided to exclude this segment from the introduction and throughout the text. As well, some phrases were added to some sections as specified bellow:
In the ABSTRACT:
“The consumption of caffeine has been suggested primarily because it improves exercise perfor-mance, reduces fatigue, and increases wakefulness and awareness Was added (page 1 lines 21-22)
“potential” was added (page 1, line 25)
“molecular docking” Was added (page 1 lines 26)
“certain mechanisms of action could be beneficial in SARS-CoV-2.” Added (page 1, line 30)
“are required for validation through in vitro and in vivo models.” Added (page 1, line 31)
In INTRODUCTION:
Page 2, second paragraph, lines 75 and page 3 lines 107 of the original manuscript were crossed out.
Page 3 second paragraph lines 131-132, we added “patients with COVID-19, especially since caffeine is consumed daily.”
In IMMUNOMODULATORY EFFECTS OF CAFFEINE:
Page 6, line 230 “indirectly” was added.
In Figure 2, we crossed out text and added “Proposed” and “of caffeine against SARS-CoV-2” in line 298.
In section CAFFEINE MIGHT BE AN INHIBITOR OF THE RBD-ACE2 COMPLEX:
Page 8, fourth paragraph, lines 349-354, we added the text: “Considering these results, the potential for the application of caffeine, both alone and in combination with antiviral agents, can be noticed. Even more so since Tong et al., have reported that ribavirin did not provide a survival benefit in comparison with the control treatment (involving only supportive therapy). Its in vivo activity against SARS-CoV-2 requires further investigations [95].
In CONCLUSIONS:
Page 9, first paragraph, lines 381 and 382, respectively text was erased, and “possible” and “both directly and indirectly, in the SARS-CoV-2 infection” were added.
REFERENCES
Tong S, Su Y, Yu Y, Wu C, Chen J, Wang S, Jiang J. Ribavirin therapy for severe COVID-19: a retrospective cohort study. Int J Antimicrob Agents. 2020 Sep;56(3):106114. doi: 10.1016/j.ijantimicag.2020.106114. Epub 2020 Jul 23. PMID: 32712334; PMCID: PMC7377772.
Reviewer 3 Report
This review manuscript aims to postulate the use of caffeine as a therapy against COVID-19.
The main premise is that caffeine can inhibit deleterious processes that are activated in COVID-19.
1-A major concern here is that there is no evidence suggesting that these mechanisms are dysregulated in COVID-19. These include the adenosine receptors, the type 2 taste receptors or even phophodiesteratses.
2-The authors suggest that caffeine may have protective effects by inhibiting adenosine receptors. However, they seem unaware of the extensive body of work highlighting the the protective effect of adenosine and its receptors in acute lung injury, an important consequence of severe COVID-19. Thus caffeine may have a detrimental effect here.
3- The manuscript also focuses extensively on airway smooth muscle, yet it is unclear whether this is a direct target for SARS-COV2.
4-Given the purported beneficial effects of caffeine, it is surprising that no studies have shown a potential difference in COVID-19 mortality of SARS-COV2 transmission in populations with higher caffeine intake.
Author Response
This review manuscript aims to postulate the use of caffeine as a therapy against COVID-19.
The main premise is that caffeine can inhibit deleterious processes that are activated in COVID-19.
- A major concern here is that there is no evidence suggesting that these mechanisms are dysregulated in COVID-19. These include the adenosine receptors, the type 2 taste receptors or even phophodiesteratses.
Answer: Thank you for your comments
In this regard, we would like to mention the following: Disease severity and mortality in COVID-19 have been linked to the development of an hyperinflammatory syndrome. Among the contributing factors in the inflammatory response, we can find an increment in the recruitment of monocytes, macrophages, somatic cells, neutrophils, NK cells, and CD4+ and CD8+ T cells in the lung parenchyma (Gustine, 2020; Rowaiye, 2021). The proinflammatory state produces the characteristic “cytokine storm,” a major player in the pathogenesis of COVID-19, where an overexpression of cytokines is observed, such as IL-6, IL-1β, TNF-α, IL-3, GM-CSF (Gustine, 2020; Rowaiye, 2021). The cytokine storm is one of the main proponents leading to acute lung injury, acute respiratory distress syndrome and multiple organ failure (Prompetchara, 2020; Gustine, 2020).
Among the many possible benefits caffeine might indirectly have in the infection caused by SARS-CoV-2, a strong emphasis should be made on the immunomodulatory effects it possesses. We are not suggesting that the signaling pathways for the adenosine receptor and TASR are dysregulated in COVID-19, but that the activity of caffeine through these mechanisms could mitigate the proinflammatory state present in COVID-19 patients.
For instance, in this manuscript we mention that TAS2R agonists, such as caffeine, could inhibit allergen-induced airway inflammation, inhibit LPS-induced cytokine release, remodeling and hyper-responsiveness (Grassin-Delyle, 2013; Nayak, 2019). Caffeine, being a known agonist of these receptors, could participate in accelerating the recovery of the sense of smell and could potentially contribute to mitigate the severity of the infection by SARS-CoV-2 due to the immunomodulatory effects attributed to these receptors (Lin, 2020; Devillier, 2015).
Meanwhile, through its antagonism of A1R, caffeine inhibits its pro-inflammatory functions in neutrophils, and eosinophils, inhibits monocyte phagocytosis, dendritic cell chemotaxis and mucus production. By antagonizing the A2BR it inhibits mast cells degranulation, cytokine secretion and IgE synthesis, in epithelial cells it inhibits IL-19 secretion; VEGF and IL-8 secretion is inhibited in endothelial cells, and IL-16 secretion from bronchial smooth muscle and fibroblasts (Brown, 2008). Based on the available data regarding caffeine anti-inflammatory capabilities, authors propose that its administration during sever pulmonary inflammatory states as Covid-19 would prove to be beneficial rather than deleterious although further and more detailed studies are needed.
We also want to comment on the recent findings regarding Sildenafil, an approved PDE-5 inhibitor used in the treatment of erectile dysfunction and pulmonary hypertension (Travadi, 2003) This drug improves pulmonary hemodynamics by reducing vascular resistance in idiopathic pulmonary fibrosis (Rochwerg, 2016; Prasad, 2000), and has proven to be effective in the treatment of COVID-19 (Sansone, 2021). We believe that caffeine, through its PDEs inhibitory capacity, could potentially have similar effects; and have therefore added some lines in this regard in the amended text. Page 4, lines 188-193.
Finally, we emphasize this is a theory that requires more research and hope that the present paper in convincing enough to grant further and better investigations.
REFERENCES
- Gustine JN, Jones D. Immunopathology of Hyperinflammation in COVID-19. Am J Pathol. 2021 Jan;191(1):4-17. doi: 10.1016/j.ajpath.2020.08.009. Epub 2020 Sep 11. PMID: 32919977; PMCID: PMC7484812.
- Rowaiye AB, Okpalefe OA, Onuh Adejoke O, Ogidigo JO, Hannah Oladipo O, Ogu AC, Oli AN, Olofinase S, Onyekwere O, Rabiu Abubakar A, Jahan D, Islam S, Dutta S, Haque M. Attenuating the Effects of Novel COVID-19 (SARS-CoV-2) Infection-Induced Cytokine Storm and the Implications. J Inflamm Res. 2021 Apr 16;14:1487-1510. doi: 10.2147/JIR.S301784. PMID: 33889008; PMCID: PMC8057798.
- Prompetchara E, Ketloy C, Palaga T. Immune responses in COVID-19 and potential vaccines: Lessons learned from SARS and MERS epidemic. Asian Pac J Allergy Immunol. 2020;38(1):1–9. doi:10.12932/AP-200220-0772
- Grassin-Delyle, S., et al., The expression and relaxant effect of bitter taste receptors in human bronchi. Respir Res, 2013. 14(1): p. 134.
- Nayak, A.P., D. Villalba, and D.A. Deshpande, Bitter Taste Receptors: an Answer to Comprehensive Asthma Control? Curr Allergy Asthma Rep, 2019. 19(10): p. 48.
- Lin, L., et al., Hypothesis for potential pathogenesis of SARS-CoV-2 infection-a review of immune changes in patients with viral pneumonia. Emerg Microbes Infect, 2020. 9(1): p. 727-732.
- Devillier, P., E. Naline, and S. Grassin-Delyle, The pharmacology of bitter taste receptors and their role in human airways. Pharmacol Ther, 2015. 155: p. 11-21.
- Brown, R.A., D. Spina, and C.P. Page, Adenosine receptors and asthma. Br J Pharmacol, 2008. 153 Suppl 1(Suppl 1): p. S446-56.
- Travadi JN, Patole SK. Phosphodiesterase inhibitors for persistent pulmonary hypertension of the newborn:A review. Pediatr Pulmonol. 2003;36:529–35.
- Liu PP, Blet A, Smyth D, Li H. The science underlying COVID-19: implications for the cardiovascular system. Circulation. 2020 doi: 10.1161/CIRCULATIONAHA.120.047549.
- Rochwerg B, Neupane B, Zhang Y, Garcia CC, Raghu G, Richeldi L, Brozek J, Beyene J, Schunemann H. Treatment of idiopathic pulmonary fibrosis: a network meta-analysis. BMC Med. 2016;14(1):18. doi: 10.1186/s12916-016-0558-x.
- Prasad S, Wilkinson J, Gatzoulis MA. Sildenafil in primary pulmonary hypertension. N Engl J Med. 2000;343(18):1342–1342. doi: 10.1056/nejm200011023431814.
- Sansone, A., Mollaioli, D., Ciocca, G. et al. Addressing male sexual and reproductive health in the wake of COVID-19 outbreak. J Endocrinol Invest 44, 223–231 (2021). https://doi.org/10.1007/s40618-020-01350-1
- The authors suggest that caffeine may have protective effects by inhibiting adenosine receptors. However, they seem unaware of the extensive body of work highlighting the the protective effect of adenosine and its receptors in acute lung injury, an important consequence of severe COVID-19. Thus caffeine may have a detrimental effect here.
Answer: Thank you for your comments
Indeed, adenosine plays a key role in regulating pulmonary inflammation and repair (Karmouty-Quintana, 2013; ter Horst, 2004). Remarkably, in acute inflammation, it significantly lowers cytokine production, immune cell migration and vascular permeability, while during chronic inflammation, continuous, sustained adenosine stimulation increases cytokine secretion and immune cell migration to the lung (Li,2016). Interestingly, caffeine administration has been shown to partially revert the consequences of persistent adenosine stimulation (Chavez-Valdez, 2009) adding another interesting feature to caffeine anti-inflammatory spectrum (Endesfelder, 2020). We find this to be a very interesting fact and are grateful to the referee that pointed out its importance. A few words in this regard were added to the amended text (page 6, Lines 254-261).
No evidence as of yet has been reported as to which adenosine receptor subtypes are predominantly activated in COVID-19, and it might depend on the stage of infection of the patient. Caffeine could mitigate lung injury mainly through antagonism of A2BR, yet further studies are required to indicate in which stage of the infection it could have a more significant protective effect.
REFRENCES
- Karmouty-Quintana H., Xia Y., Blackburn M. R. Adenosine signaling during acute and chronic disease states. Journal of Molecular Medicine. 2013;91(2):173–181. doi: 10.1007/s00109-013-0997-1.
- ter Horst SA, Wagenaar GT, de Boer E, van Gastelen MA, Meijers JC, Biemond BJ, Poorthuis BJ, Walther FJ. Pentoxifylline reduces fibrin deposition and prolongs survival in neonatal hyperoxic lung injury. J Appl Physiol (1985). 2004 Nov;97(5):2014-9. doi: 10.1152/japplphysiol.00452.2004. Epub 2004 Jun 18. PMID: 15208286.
- Li H., Karmouty-Quintana H., Chen N. Y., et al. Loss of CD73-mediated extracellular adenosine production exacerbates inflammation and abnormal alveolar development in newborn mice exposed to prolonged hyperoxia. Pediatric Research. 2017;82(6):1039–1047. doi: 10.1038/pr.2017.176.
- Chavez-Valdez R, Wills-Karp M, Ahlawat R, Cristofalo EA, Nathan A, Gauda EB. Caffeine modulates TNF-alpha production by cord blood monocytes: the role of adenosine receptors. Pediatr Res. 2009 Feb;65(2):203-8. doi: 10.1203/PDR.0b013e31818d66b1. PMID: 19047957.
- Endesfelder S, Strauß E, Bendix I, Schmitz T, Bührer C. Prevention of Oxygen-Induced Inflammatory Lung Injury by Caffeine in Neonatal Rats. Oxid Med Cell Longev. 2020 Aug 7;2020:3840124. doi: 10.1155/2020/3840124. PMID: 32831996; PMCID: PMC7429812.
- The manuscript also focuses extensively on airway smooth muscle, yet it is unclear whether this is a direct target for SARS-COV2.
Answer: Thank you for your comments
Recently, Kalidhindi et al., using confocal imaging, found that ACE2 is expressed in human airway smooth muscle (ASM), although an entry of SARS-CoV-2 has not been demonstrated. The importance of these cells lies in the possibility that SARS-CoV-2 targets and damages the epithelial layers with subsequent influences on the underlying mesenchymal cells [ASM, fibroblasts], which leads to an alteration in the reactivity of the airways, inflammation and long-term fibrosis. In this context, SARS-CoV-2 can directly influence the functionality of multiple signaling pathways in lung cells, including ASM.
On the other hand, a pro-inflammatory environment such as that described in patients with Covid-19 that causes inflammation through monocytes, macrophages, neutrophils, NK cells, CD4 and CD8 + T cells, (Gustine et al, 2020) induces the production of cytokines that cause hyperresponsiveness and remodeling of the airways. In this context, we believe that caffeine can negatively regulate all of these processes and benefit ASM homeostasis. Additionally, caffeine has been shown to improve asthma in adults. People with mild to moderate asthma improved their lung function (Welsh et al., 2010). Caffeine also showed a bronchodilator effect in young asthma patients (Becker et al., 1984).
A few words in this regard were added to the amended text (page 3, Lines 135-141).
REFRENCES
- Kalidhindi RSR, Borkar NA, Ambhore NS, Pabelick CM, Prakash YS, Sathish V. Sex steroids skew ACE2 expression in human airway: a contributing factor to sex differences in COVID-19? Am J Physiol Lung Cell Mol Physiol. 2020 Nov 1;319(5):L843-L847. doi: 10.1152/ajplung.00391.2020. Epub 2020 Sep 30. PMID: 32996784; PMCID: PMC7789973
- Gustine JN, Jones D. Immunopathology of Hyperinflammation in COVID-19. Am J Pathol. 2021 Jan;191(1):4-17. doi: 10.1016/j.ajpath.2020.08.009. Epub 2020 Sep 11. PMID: 32919977; PMCID: PMC7484812.
- Welsh EJ, Bara A, Barley E, Cates CJ. Caffeine for asthma. Cochrane Database of Systematic Reviews 2010, Issue 1. Art. No.: CD001112. DOI: 10.1002/14651858.CD001112.pub2. Accessed 24 April 2021.
- Becker, A.B., Simons, K.J., Gillespie, C.A., Simons, F.E., 1984. The bronchodilator effects and pharmacokinetics of caffeine in asthma. N. Engl. J. Med. 310, 743–746.
- Given the purported beneficial effects of caffeine, it is surprising that no studies have shown a potential difference in COVID-19 mortality of SARS-COV2 transmission in populations with higher caffeine intake.
Answer: Thank you for your observation, we appreciate your fine comment!
We had definitively not thought about the correlation between a higher habitual caffeine intake and mortality of SARS-CoV-2. It is a fascinating question that merits further in-depth study. Moved by curiosity, we did a comparison between Finland, the country with the highest coffee consumption (Kg consumed by a person/year) as an example of caffeine consumption and the United States that holds the 20th place in coffee consumption worldwide, and the mortality reported due to SARS-CoV-2 (Table 1). Notably, Finland data reveal a better general state of health that might hinder COVID-19 infection from the beginning as severity and mortality in SARS-CoV-2 is multifactorial, with various risk factors playing a role like hypertension, diabetes mellitus and obesity.
- Institute for Health Metrics and Evaluation (IHME). GBD Compare. Seattle, WA: IHME, University of Washington, 2015. Available from http://vizhub.healthdata.org/gbd-compare
- Dong E, Du H., Gardner L. An interactive web-based dashboard to track COVID-19 in real time. The Lancet infectious disease. 2020. 20,5, P533-534.
We do not wish to speculate that a higher coffee consumption could impact mortality due to SARS-CoV-2; however, Finland, the 1st coffee consuming nation, has a lower mortality than the United States, and this observation merits further study to see if caffeine consumption could be correlated with the mortality rate in SARS-CoV-2.
Caffeine might also indirectly impact the likelihood of contracting a moderate or severe case of COVID-19 infection by ameliorating the risk of developing some comorbidities that have been associated with COVID-19 unfavorable prognosis (mainly hypertension and diabetes). Caffeine has well recognized cardioprotective qualities, with significant risk reduction of cardiovascular events with every cup consumed. In comparison with none coffee drinkers, every cup a day reduced the risk of heart failure by 7%, stroke by 8% and coronary heart disease by 5% (O’keefe, 2018; Mostofsky, 2012). Furthermore, a negative correlation has been noticed between coffee consumption and the likelihood to develop Type 2 Diabetes Mellitus, where each additional cup consumed a day reduced the risk by 7% (Ding, 2014; Huxley, 2009). Seemingly, this alkaloid possesses promising therapeutic potential that warrants further and more extensive research. We would like to ask the referee if he/she considers some lines in this regard should be included in the manuscript.
Authors are grateful to the Editor and Referees for their comments and observations and are hopeful that with the amendments described herein our manuscript might be publishable in your distinguished journal.
REFRENCES
- Singh AK, Gillies CL, Singh R, Singh A, Chudasama Y, Coles B, Seidu S, Zaccardi F, Davies MJ, Khunti K. Prevalence of co-morbidities and their association with mortality in patients with COVID-19: A systematic review and meta-analysis. Diabetes Obes Metab. 2020 Oct;22(10):1915-1924. doi: 10.1111/dom.14124. Epub 2020 Jul 16. PMID: 32573903; PMCID: PMC7361304.
- O’Keefe, J. H., DiNicolantonio, J. J., & Lavie, C. J. (2018). Coffee for Cardioprotection and Longevity. Progress in Cardiovascular Diseases, 61(1), 38–42. doi:10.1016/j.pcad.2018.02.002
- Mostofsky E, Rice MS, Levitan EB, Mittleman MA. Habitual coffee consumption and risk of heart failure: a dose-response meta-analysis. Circ Heart Fail. 2012; 5:401-405.
- Steffen M, Kuhle C, Hensrud D, Erwin PJ, Murad MH. The effect of coffee consumption on blood pressure and the development of hypertension: a systematic review and meta-analysis. J Hypertens. 2012;30:2245-2254.
- Ding M, Bhupathiraju SN, Chen M, van Dam RM, Hu FB. Caffeinated and decaffeinated coffee consumption and risk of type 2 diabetes: A systematic review and dose response meta-analysis. Diabetes Care. 2014;37:569-586.
- Huxley R, Lee CM, Barzi F, Timmermeister L, Czernichow S, Perkovic V, Grobbee DE, Batty D, Woodward M. Coffee, decaffeinated coffee, and tea consumption in relation to incident type 2 diabetes mellitus: a systematic review with meta-analysis. Arch Intern Med. 2009 Dec 14;169(22):2053-63. doi: 10.1001/archinternmed.2009.439. PMID: 20008687
